# Cardiometabolic Risk in a University Community: An Observational Study

**DOI:** 10.3390/healthcare12171756

**Published:** 2024-09-03

**Authors:** David Pérez-Manchón, Jaime Barrio-Cortes, Angel Vicario-Merino, Noemí Mayoral-Gonzalo, Montserrat Ruiz-López, Eduardo Corral-Pugnaire, Patricia Blanco-Hermo, Cayetana Ruiz-Zaldibar

**Affiliations:** 1Department of Nursing, Faculty HM of Health Sciences, University Camilo Jose Cela, 28692 Madrid, Spain; dpmanchon@ucjc.edu (D.P.-M.); avicario@ucjc.edu (A.V.-M.); nmayoral@ucjc.edu (N.M.-G.); mrlopez@ucjc.edu (M.R.-L.); ejcorral@ucjc.edu (E.C.-P.); pblanco@ucjc.edu (P.B.-H.); crzaldibar@ucjc.edu (C.R.-Z.); 2Health Research Institute of HM Hospitals, 28015 Madrid, Spain

**Keywords:** cardiometabolic risk, university personnel, health promotion, nursing

## Abstract

The highest prevalence of cardiovascular risk factors has been associated with obesity, sedentary lifestyle, and elevated blood pressure due to high workload and work stress. This study aimed to analyze the cardiometabolic risk and lifestyles among the health sciences university academics and campus administrators at a private university in Spain. A cross-sectional study was conducted during the 2018–2019 academic year by the Nursing Department, using a self-administered questionnaire and face-to-face assessments of anthropometric variables related to cardiovascular risk in university personnel. The variables measured included sociodemographics, cardiovascular risk history, comorbidities, toxic habits, Mediterranean diet adherence, physical exercise, psychosocial stress, and physical, anthropometric, and analytical data. Cardiovascular risk was categorized into relative (<40 years), absolute, and vascular age (>40 years). Among the 101 participants, 61.4% were women, with a mean age of 41.3 years ± 9 years. The smoking prevalence was 21.8% (68.2% women), 27.7% were sedentary, and 51.0% adhered to the Mediterranean diet, with higher adherence among the academics. Emotional risk was present in 32.7% of the participants. A prior diagnosis of hypertension was significantly more frequent in the men (15.4%) compared to the women (3.2%). The blood pressure measurements were mostly optimal across both genders and professional groups, but the proportion of hypertension grade 1 was significantly higher among the academics (10%) compared to the administrators (4.5%) and among the men (11.1%) compared to the women (5.9%). The absolute cardiovascular risk among the university employees was generally low, but the men exhibited a more moderate risk compared to the women. It is necessary for the university to promote health within its community, with the Nursing Department playing a key role in health promotion and research.

## 1. Introduction

Cardiovascular diseases (CVDs) continue to be a major health problem. Most population-based cardiovascular events are non-fatal, but, among the fatal events, 80% appear in the low and moderate risk categories, with no gender differences [1]. The origin lies within the general context of arteriosclerosis as a proinflammatory and prothrombotic process. Cardiovascular risk (CVR) and cardiometabolic risk (CMR) estimate the probability of these events occurring. At the population level, the evidence recommends evaluating the relative CVR in people under 40 years and absolute CVR in those over 40 years using the SCORE system [2], which estimates the 10-year cumulative risk of a first fatal atherosclerotic event, as well as estimating the vascular age or age of the arteries [3].

The main intervention strategy in CVDs should be primary prevention through the detection of cardiovascular risk factors (CVRFs) [4]. The current recommendations [5] include assessing aspects related to individual health and lifestyles. However, continuous population changes, high labor and emotional demands, and technological advances make it necessary to also assess the work environment. This environment is usually sedentary [6], as is the case for the university setting [7]. The highest prevalence of CVRFs in the university community is associated with obesity, sedentary lifestyle, and elevated blood pressure [8,9]. These factors, combined with the demanding workdays in academia, work-related stressors from teaching and research, and university lifestyles, can increase CVR from a young age.

The aim of this investigation was to study the cardiovascular and cardiometabolic risks, as well as lifestyles, among the university academics (UAs) and university campus administrators (UCAs) at the University Camilo José Cela (UCJC) in Madrid, Spain.

## 2. Materials and Methods

### 2.1. Design, Setting and Study Subjects

A cross-sectional study was conducted by the UCJC Nursing Department between October 2018 and May 2019 on the university campus, utilizing a self-administered online questionnaire and face-to-face assessments of physical and anthropometric variables related to CVR in university personnel.

The recruitment for the study was completed using an invitation through the institutional mail, with signs and using the snowball effect throughout the UCJC campus. The target population was around 400 university professionals.

The inclusion criteria consisted of the UAs from the Faculty of Health Sciences and all the UCAs at UCJC, with either full-time, part-time dedication, or guest lectures. The exclusion criteria involved individuals with congenital heart defects (such as atrial septal defect, ventricular septal defect, patent ductus arteriosus, tetralogy of Fallot, pulmonary stenosis, aortic stenosis, transposition of the great arteries, tricuspid atresia, coarctation of the aorta, total anomalous pulmonary venous return, hypoplastic left heart syndrome, Ebstein’s anomaly, single ventricle defect, double outlet right ventricle, and mitral valve prolapse) or established CVD (such as coronary artery disease, heart failure, atrial fibrillation, myocardial infarction, stroke, peripheral artery disease, aortic aneurysm, valvular heart disease, cardiomyopathy, myocarditis, and endocarditis) to high CVR, as well as university maintenance personnel. Sampling was non-probability, successive, voluntary, and consecutive, provided that the participants met the inclusion criteria.

Staff interested in participating in the study were instructed to schedule an appointment, which was confirmed by the researchers. Upon arrival at the appointment, the researcher reviewed and obtained the participant’s signature on the Informed Consent form. The researcher then collected all required information, completed the survey, and addressed any questions or concerns the participant might have.

### 2.2. Data Collection

Recruitment was conducted through the university’s internal information channels, using online consent for the electronic questionnaire and a voluntary registration link to schedule face-to-face assessments at specific dates and times.

### 2.3. Variables

The measurements included sociodemographic and work-related variables (full-time positions, dedication involves approximately 40 h per week; part-time position involves around 20 h per week; and guest lectures are measured based on the number of lecture sessions), family and personal history of CVR, presence of comorbidities with current pharmacological treatment, and lifestyle factors related to toxic habits.

Adherence to the Mediterranean diet was assessed using the DietMed questionnaire [10], while physical activity was evaluated using the IPAQ questionnaire [11]. Psychosocial stress was measured with the GHQ-12 scale [12].

Physical measurements included blood pressure (mmHg) and heart rate (beats per minute) using a validated electronic blood pressure monitor, while anthropometric measurements included weight (kg), height (m^2^), Body Mass Index (kg/m^2^), and abdominal circumference (centimeters). Analytical measurements consisted of glucose and lipid profiles (mg/dL) obtained through capillary blood (participants were informed to maintain standardized fasting conditions for measurement when their appointment was confirmed). The lipid profile analyzer [13] met international precision and accuracy standards [14].

### 2.4. Statistical Analysis

Statistical analysis was performed using the IBM SPSS Statistics 21.0 software. Quantitative variables were presented as means and standard deviations; non-parametric tests were used for non-normal distributions. The Shapiro–Wilk test was used to assess normality. Qualitative variables were reported with absolute and relative values. For bivariate analysis, the Student’s *t*-test was used to compare means, Fisher’s exact test was applied for small sample sizes, and chi-squared tests were used to investigate differences in proportions between qualitative variables. The Bonferroni method was used for multiple comparisons. To define risk profiles in the UAs and UCAs by age, multivariate analysis with binary logistic regression was conducted. The *p*-value considered statistically significant was *p* < 0.05.

## 3. Results

A total of 101 participants completed the self-administered online questionnaire, comprising 38.6% men and 61.4% women. The mean age was 41.3 years, with 66.3% being UAs and 33.7% being UCAs. Additionally, 71.3% worked full-time at the university, while 26.7% were part-time associate professors, and 2% were guest lecturers. Non-statistically significant differences were found regarding gender, age, distribution, and employment status when comparing the employed groups, as shown in Table 1.

When considering the CVRFs (Table 2), 21.8% of the participants were smokers, with a higher proportion of the women compared to the men (68.2% vs. 31.8%), as well as among the UAs (86.4%) and those with exclusive work dedication at the university (86.4%). The average consumption was 8–9 cigarettes per day, being higher in the UCAs (9 cigarettes/day) compared to the UAs (7.9 cigarettes/day). However, none of these differences regarding gender or employed status were statistically significant. Of the remaining participants, 27.7% were ex-smokers and 50.5% were non-smokers. The adherence to the Mediterranean diet was 54.0% overall, with similar adherence rates among the males (48.7%) and females (51.6%), but with statistically significantly greater adherence in the UAs (59.7%) than in the UCAs (32.4%). Additionally, 27.7% were sedentary or had low physical activity, averaging 6–7 h of seated work per day. Among the rest, 39.6% practiced vigorous weekly physical activity, while 32.7% engaged in moderate activity. Regarding psychosocial stress, 32.7% exhibited emotional risk. In relation to the obesity level, 15.8% had overweight and 6.9% obesity. However, no statistically significant differences were found in these variables with respect to gender or employment status. Regarding the blood pressure measured in the study, it was mostly optimal across both the gender and professional groups, but the category of grade 1 hypertension had a higher proportion in the UAs (10%) compared to the UCAs (4.5%) and in the men compared to the women, with these differences also being statistically significant.

Regarding CVR, as shown in Table 3, 91.1% of the participants had a family history of CVR, 7.9% had hypertension, 5% had sleep apnea syndrome, 10.9% had hypercholesterolemia, and 26.7% had hypertriglyceridemia. A prior diagnosis of hypertension was significantly higher in the men (15.4%) compared to the women (3.2%). No other statistically significant differences were found for these comorbidities with respect to gender or employment status. The CVR by comorbidities was assessed in 52 participants (51.5% of the total sample). It was disaggregated into relative CVR for individuals under 40 years (26.7%) and total CVR with vascular age for those over 40 years (23.8%). Among those evaluated, the relative and absolute CVR values were mostly low (70.4% and 87.5%), and 37.6% had added risk due to associated comorbidities (14.3% cardiac, 21.4% systemic inflammatory, and 21.4% endocrine–metabolic). There were statistically significant differences in the estimation of the absolute cardiovascular risk by gender, with higher risk in the men compared to the women, but similar risk across the professional groups.

In terms of the vascular age calculated, 5.9% had a vascular age lower than their chronological age, 3% had the same age, and 14% had a higher vascular age. No statistically significant differences were found when comparing the gender and employment groups.

## 4. Discussion

The university must act as a leader in health promotion within the community [15] by fostering healthy university networks [16]. Characterizing the lifestyles and the cardiometabolic profile of the university community is essential for developing strategies to promote health and prevent disease among the students, UAs, and UCAs. However, there is limited research on the determinants of university health and their impact on cardiovascular risk. The work environment plays a fundamental role, and it is crucial to promote cardiovascular health through interventions aimed at improving lifestyles, emotional well-being, and reducing cardiometabolic risk factors [17]. This approach not only benefits health outcomes but also fosters a productive workforce [18].

In this context, the nursing professionals working in the university are key factors, providing essential leadership and expertise in implementing health promotion strategies and supporting the overall well-being of the university community while also being vital in research to advance healthcare by generating evidence that informs the practice and improves the patient outcomes [19,20,21].

An international study warned of a high prevalence of CVRFs among teaching employees and university campus researchers in recent years, particularly concerning overweight, high blood pressure, and prediabetes, suggesting increased risks of obesity, type 2 diabetes, and hypertension [22]. In the Spanish university setting, the most frequent CVRF is obesity, with higher rates in women than in men, followed by overweight, which shows an inverse relationship by gender [23]. In our study, a significant portion of the participants were classified as overweight, with a notable percentage showing obesity and abdominal obesity, indicating medium effect sizes in both abdominal and BMI obesity for the university academics and campus administrators. However, these findings are slightly lower compared to other population studies [24], which may be attributed to the protective effect of a reasonably high adherence to the Mediterranean diet within the context of the university food services.

In our research, the most prevalent CVRF was smoking, with a significant portion of the participants identified as smokers, particularly among the women, whose prevalence was double that of the men. Notably, the smoking profile was primarily associated with those researchers dedicated exclusively to the university, who reported emotional stressors and higher daily cigarette consumption, especially if they were also sedentary. Although these findings were not statistically significant, they align with other studies [23] and reflect the trends in the broader Spanish population [25].

In relation to other unhealthy lifestyles, our study found that almost a third of the participants were sedentary, spending an average of 6 to 7 h a day seated, which is slightly lower than the findings from similar studies [22,26]. Although the higher proportion of sedentary individuals and abdominal obesity among the females did not reach statistical significance, it could still contribute to an increased cardiometabolic risk [27]. A sedentary lifestyle may significantly impact the health of the university population due to the extensive hours spent in sedentary work. Implementing physical activity interventions within the university work environment could be a cost-effective strategy to enhance the health of faculty and researchers, potentially reducing absenteeism and improving productivity [28,29,30,31].

The emotional factors related to the work environment, particularly affecting women and university academics, may also contribute to this increased risk, as shown in other university communities [32,33].

Studies indicate that job stressors and long working hours are associated with a moderate increase in the risk of coronary events, strokes, and even type 2 diabetes [34]. In our study, a notable proportion of the participants reported emotional risk, with higher rates observed among the women compared to the men and among the university academics compared to the university campus administrators. Despite the lack of statistical significance in these differences, the importance and impact on university health have led to the implementation of campaigns aimed at preventing occupational stress and enhancing health among teachers, researchers, and other staff members. These initiatives are a vital component of health promotion in the university environment.

Differences were also identified in the cardiovascular health of the teaching and research staff compared to the service staff. The UCAs exhibited an increased prevalence of smoking, sedentary lifestyles, and higher rates of overweight and abdominal obesity, along with significantly lower adherence to the Mediterranean diet. However, they reported lower daily cigarette consumption along with a lower proportion of class I obesity. Except for the greater adherence to the Mediterranean diet in the UAs, these differences were not statistically significant. However, a larger effect size was observed in the association between gender and physical activity compared to the association between UCJC employee status and physical activity, indicating that physical activity is more strongly related to gender than to employee status. These differences may stem from the greater work dedication of university campus administrators, reflected in their longer hours according to their employment contracts, as well as higher emotional stress due to their workloads. This situation likely results in increased presence on campus and, consequently, less healthy lifestyle choices.

The combination of the described risk factors and unhealthy lifestyles may contribute to increased cardiovascular risk. However, given the relatively young average age of the study population, this could act as a protective factor. Most of the participants exhibited low cardiovascular risk, both globally and in absolute terms for those over 40, but the men exhibited a more moderate absolute CVR compared to the women. Although a significant portion of the participants had a vascular age several years higher than their chronological age, and some had additional cardiovascular risk due to comorbidities, these findings were not statistically significant. Therefore, caution is warranted when interpreting the impact of these factors on the overall cardiovascular risk.

The university environment in which the UCAs and UAs carry out their work is an ideal setting for promoting lifestyle changes given their high daily work commitment. Strategies such as increasing daily steps and encouraging walking among university staff have been linked to reductions in blood pressure and abdominal circumference [35,36]. However, there is limited knowledge regarding the effectiveness of these programs within the university population. Despite the university setting being conducive to health promotion [26], the rising demands and stressors faced by the employees in this sector have been associated with a high prevalence of cardiovascular events [37].

Nevertheless, the study is not without limitations. It was conducted during the 2018–2019 academic year, but, despite that, it provides a comprehensive assessment of the cardiovascular risk factors among the university personnel. The COVID-19 pandemic has significantly impacted cardiovascular disease risk factor prevention and management, particularly in disadvantaged communities such as older adults, racial and ethnic minority groups, and those facing adverse socioeconomic circumstances [38], different from our study population. Even so, our findings suggest that the cardiovascular risk profiles among university populations have remained consistent when compared with the post-pandemic data [39,40]. Additionally, national studies of working-age adults indicate that, while routine screenings have not yet returned to the pre-pandemic levels, risk factor treatment has remained stable [41]. Therefore, these data are still important for understanding and addressing the cardiovascular risk within this specific demographic and underscore the importance of detailed assessments despite the inherent difficulties and evolving circumstances. Also, this study included a small sample size, which may affect the statistical significance of the differences observed. The consecutive sampling method might not fully represent the entire university community, and there could be potential biases due to voluntary participation, classification bias in the administered questionnaires, and intra-observer error among the participating researchers. Moreover, cardiovascular risk was assessed in only a subset of the participants. However, while it focuses on identifying a cardiometabolic profile among the university personnel at a local level, the findings provide valuable insights when compared with personnel from other universities, both within the same country and internationally. Furthermore, the study features a comprehensive assessment of various CVRFs, utilizes validated questionnaires, and involves nursing researchers within a multidisciplinary team for data collection and health promotion recommendations. This collaborative approach not only enhances the robustness and relevance of the findings within the university context but also underscores the critical role of nursing in community health promotion and research.

## 5. Conclusions

The total CVR among the university employees in this university was generally low, but the men exhibited more moderate levels of risk compared to the women. The most prevalent unhealthy lifestyle factors were physical inactivity and smoking. Adherence to the Mediterranean diet was observed in only half of the sample, with notably higher adherence in the UAs compared to the UCAs.

Although the factors typically associated with elevated CVR, such as blood pressure levels, were mostly optimal, higher rates of grade I hypertension were noted in the men compared to the women and in the UAs compared to the UCAs. Nearly all the participants had a family history of cardiovascular risk (CVR). A small proportion had a previous diagnosis of hypertension (with higher rates in the men), a few had sleep apnea syndrome, some had hypercholesterolemia, and a notable fraction had hypertriglyceridemia. Additionally, more than one in five participants were affected by overweight or obesity.

These findings underscore the need for universities to act as leaders in health promotion, using their setting to implement lifestyle changes and foster community engagement through healthy university networks to reduce the prevalence of cardiovascular events. The current study establishes an important foundation and highlights the need for further research in this area. Future studies can build on this work to provide a more comprehensive analysis, but the present findings already contribute significantly to the understanding of cardiovascular risk in university personnel.

Additionally, the involvement of the nursing staff is crucial as they play a vital role in implementing health promotion strategies, providing support, and encouraging healthier behaviors within the university community. Nursing professionals are also essential in research efforts, contributing valuable insights and evidence-based practices that enhance the understanding and improvement of health outcomes in academic settings. Addressing these gaps and leveraging the expertise of their nursing staff can enable universities to make significant strides in promoting cardiovascular health.

## Figures and Tables

**Table 1 healthcare-12-01756-t001:** Sociodemographic and work-related variables.

Variable Number (%)	Total 101 (100)	University Campus Administrators 34 (33.7)	University Academics 67 (66.3)	Test Statistics *p*-Value
**Gender**				
Males	39 (38.6)	7 (6.9)	32 (31.7)	χ^2^ = 7.026 * *p* = 0.723
Females	62 (61.4)	27 (26.7)	35 (34.7)	
**Age ^a^**	41.3 years ± 9.4	39.2 years ± 9.3	42.4 years ± 9.3	*p* = 0.107 **
Males	43.3 years ± 9.7	40.0 years ± 11.0	44.0 years ± 9.5	*p* = 0.335 **
Females	40.1 years ± 9.0	39.04 years ± 9.0	41.0 years ± 9.0	*p* = 0.399 **
**University time dedication**				
Full-time dedication	72 (71.3)	31 (91.2)	41 (61.2)	χ^2^ = 7.026 * *p* = 0.08
Partial-time dedication	27 (26.7)	3 (8.8)	24 (35.8)	
Guest lecturers	2 (2.0)	0 (0)	2 (3.0)	

^a^ Mean (standard deviation). * Chi-squared. ** Student’s *t*-test.

**Table 2 healthcare-12-01756-t002:** Cardiovascular risk factors.

Variable Number (%)	Total	Males	Females	Statistics	University Campus Administrators	University Academics	Statistics
**Smoking Status**	101 (100)	39 (38.5)	62 (61.4)		34 (33.7)	67 (66.3)	
Number of cigarettes ^a^	8.2 ± 6.9	9.1 ± 7.5	7.7 ± 6.7	*p* = 0.602 ***	9.0 ± 6.9	7.9 ± 7.2	*p* = 0.539 ***
Smoker	22 (21.8)	7 (17.9)	15 (24.2)	χ^2^ = 0.649 * *p* = 0.723 0.080 **	8 (23.5)	14 (20.9)	χ^2^ = 0.465 * *p* = 0.792 0.067 **
Non-smoker	51 (50.5)	20 (51.3)	31 (50)	χ^2^ = 0.649 * *p* = 0.723 0.080 **	18 (52.9)	33 (49.3)	
Ex-smoker	28 (27.7)	12 (30.8)	16 (25.8)		8 (23.5)	20 (29.9)	
**Adherence to Mediterranean diet**	101 (100)	39 (38.5)	62 (61.4)		34 (33.7)	67 (66.3)	
Yes	54 (54,5)	19 (48.7)	32 (51.6)	χ^2^ = 0.080 * *p* = 0.431 0.777 **	11 (32.4)	40 (59.7)	χ^2^ = 6.748 * ***p* = 0.009** 0.258 **
No	47 (46.5)	20 (51.3)	30 (48.4)		23 (67.6)	27 (40.3)	
**IPAQ questionnaire**	101 (100)	39 (38.5)	62 (61.4)		34 (33.7)	67 (66.3)	
Vigorous	40 (39.6)	20 (51.3)	20 (32.3)	χ^2^ = 4.528 * *p* = 0.104 0.210 **	14 (41.2)	26 (38.8)	χ^2^ = 1.032 * *p* = 0.597 0.101 **
Moderate	33 (32.7)	12 (30.8)	21 (33.9)		9 (26.5)	24 (35.8)	
Low or inactive	28 (27.7)	7 (17.9)	21 (33.9)		11 (32.4)	17 (25.4)	
**Number of hours seated ^a^**	6.7 h ± 2.5	5.6 h ± 2.5	7.3 h± 2.3	*p* = 0.07 ***	7.2 h ± 2.2	6.4 h ± 2.6	*p* = 0.264 ***
**GHQ-12**	101 (100)	39 (38.5)	62 (61.4)		34 (33.7)	67 (66.3)	
No emotional risk	68 (67.3)	28 (71.8)	40 (64.5)	χ^2^ = 0.577 * *p* = 0.448 0.076 **	23 (67.6)	45 (67.2)	χ^2^ = 0.002 * *p* = 1.00 0.005 **
With emotional risk	33 (32.7)	11 (28.2)	22 (35.5)		11 (32.4)	22 (32.8)	
**Obesity level (BMI) ^b^**	52 (51,5)	18 (34.6)	34 (65.4)		22 (42.3)	30 (57.7)	
Normal weight ^c^	29 (28.7)	10 (55.6)	19 (55.9)	χ^2^ = 2.987 * *p* = 0.560 0.228 **	13 (59.1)	16 (53.3)	χ^2^ = 5.447 * *p* = 0.244 0.324 **
Overweight class I ^d^	7 (6.9)	4 (22.2)	3 (8.8)		1 (4.5)	6 (20)	
Overweight class II ^e^	9 (8.9)	2 (11.1)	7 (20.6)		3 (13.6)	6 (20)	
Obesity class I ^f^	6 (5.9)	2 (11.1)	4 (11.8)		4 (18.2)	2 (6.7)	
Obesity class III ^g^	1 (1)	0 (0)	1 (2.9)		1 (4.5)	0 (0)	
**Abdominal Obesity**	52 (51.5)	18 (34.6)	34 (65.4)		22 (42.3)	30 (57.7)	
Yes ^h^	16 (15.8)	3 (16.7)	13 (38.2)	χ^2^ = 2.739 * *p* = 0.098 0.222 **	7 (31.8)	9 (30)	χ^2^ = 0.020 * *p* = 0.888 0.019 **
No	36 (35.6)	15 (83.3)	21 (61.8)		15 (68.2)	21 (70)	
**Blood Pressure**	52 (51.5)	18 (34.6)	34 (65.4)		22 (42.3)	30 (57.7)	
Optimal	31 (30.7)	8 (44.4)	23 (67.6)	χ^2^ = 2.769 * ***p* = 0.049** 0.232 **	12 (54.5)	19 (63.3)	χ^2^ = 7.869 * ***p* = 0.049** 0.389 **
Normal	12 (11.9)	6 (33.3)	6 (17.6)		4 (18.2)	8 (26.7)	
High normal	5 (5)	2 (11.1)	3 (8.8)		5 (22.7)	0 (0)	
Gr. 1 hypertension	4 (4)	2 (11.1)	2 (5.9)		1 (4.5)	3 (10)	

^a^ Mean (standard deviation). ^b^ Body Mass Index. ^c^ BMI:18.50–24.9 Kg/m^2^. ^d^ BMI: 25–26.9 Kg/m^2^. ^e^ BMI: 27–29.9 Kg/m^2^. ^f^ BMI: 30–34.9 Kg/m^2^. ^g^ BMI: >40 Kg/m^2^. ^h^ Males (>102 cm). Females (>88 cm). * Chi-squared. ** Effect size: Phi and Cramer’s V. *** Student’s *t*-test. BMI: Body Mass Index. CI: Confidence interval. GHQ-12: General Health Questionnaire-12. IPAQ: Physical Activity Questionnaire.

**Table 3 healthcare-12-01756-t003:** Cardiovascular risk.

Variables	Total N (%)	Males	Females	Statistics	University Campus Administrators	University Academics	Statistics
**Family Cardiovascular Risk**	101 (100)	39 (38.5)	62 (61.4)		34 (33.7)	67 (66.3)	
Yes	92 (91.1)	35 (89.7)	57 (91.9)	χ^2^ = 0.140 * *p* = 0.709 0.037 **	32 (94.1)	60 (89.6)	χ^2^ = 0.579 * *p* = 0.447 0.076 **
No	9 (8.95)	4 (10.3)	5 (8.1)		2 (5.9)	7 (10.4)	
**Hypertension**	101 (100)	39 (38.5)	62 (61.4)		34 (33.7)	67 (66.3)	
Yes	8 (7.9)	6 (15.4)	2 (3.2)	χ^2^ = 4.853 * ***p* = 0.028** 0.219 **	5 (14.7)	3 (4.5)	χ^2^ = 3.235 * *p* = 0.07 0.179 **
No	93 (92.1)	33 (84.6)	60 (96.8)		29 (85.3)	64 (95.5	
**Sleep Apnea Syndrome**	101 (100)	39 (38.5)	62 (61.4)		34 (33.7)	67 (66.3)	
Yes	5 (5)	3 (7.7)	2 (3.2)	χ^2^ = 0.982 * *p* = 0.322 0.100 **	3 (8.8)	2 (3.0)	χ^2^ = 1.526 * *p* = 0.217 0.127 **
No	96 (95)	36 (92.3)	60 (96.8)		31 (91.2)	65 (97)	
**Hypercholesterolemia**	52 (51.5)	18 (34.6)	34 (65.4)		22 (42.3)	30 (57.7)	
Yes ^a^	12 (10.9)	5 (27.8)	7 (20.6)	χ^2^ = 0.057 * *p* = 0.811 0.034 **	4 (18.2)	8 (26.7)	χ^2^ = 0.265* *p* = 0.606 0.072 **
No ^b^	40 (39.6)	13 (72.2)	27 (79.4)		18 (81.8)	22 (73.3)	
Hypertriglyceridemia	52 (51.5)	18 (34.6)	34 (65.4)		22 (42.3)	30 (57.7)	
Yes ^c^	27 (26.7)	9 (50)	19 (55.9)	χ^2^ = 0.354 * *p* = 0.552 0.083 **	13 (59.8)	15 (50)	χ^2^ = 0.587 * *p* = 0.443 0.107 **
No ^d^	24 (23.8)	9 (50)	15 (44.1)		9 (40.9)	15 (50)	
**Relative Cardiovascular Risk ^e^**	27 (26.7)	9 (33.3)	18 (66.7)		14 (51.8)	13 (48.2)	
Low Risk	19 (18.8)	6 (66.7)	13 (72.2)	χ^2^ = 0.872 * *p* = 0.647 0.144 **	3 (21.4)	8 (61.5)	χ^2^ = 2.751 * *p* = 0.253 0.295 **
Moderate Risk	8 (7.9)	3 (33.3)	5 (27.8)		11 (78.6)	5 (38.5)	
**Vascular Age ^f^**	24 (23.8)	8 (33.3)	16 (66.7)		8 (33.3)	16 (66.7)	
Lower vascular age vs. age years	6 (5.9)	2 (25)	4 (25)	χ^2^ = 10.277 * *p* = 0.506 0.274 **	2 (25)	4 (25)	χ^2^ = 8.595 * *p* = 0.659 0.292 **
Greater vascular age vs. age years	15 (14.9)	5 (62.5)	10 (62.5)		2 (25)	11 (68.8)	
Same vascular age vs. age years	3 (3)	1 (12.5)	2 (12.5)		4 (50)	1 (6.3)	
**Absolute Cardiovascular Risk ^f^**	24 (23.8)	8 (33.3)	16 (66.7)		8 (33.3)	16 (66.7)	
Low Risk	21 (20.8)	5 (62.5)	16 (100)	χ^2^ = 6.857 * ***p* = 0.009** 0.535 **	7 (87.5)	14 (87.5)	χ^2^ = 0.000 * *p* = 1.000 0.000 **
Moderate Risk	3 (3)	3 (37.5)	0 (0)		1 (12.5)	2 (12.5)	
High or very high risk	0 (0)	0 (0)	0 (0)		0 (0)	0 (0)	
**Cardiovascular Risk** **by comorbidity**	52 (51.5)	18 (34.6)	34 (65.4)		22 (42.3)	30 (57.7)	
Cardiovascular Risk added	38 (37.6)	14 (82.4)	24 (70.6)	χ^2^ = 863 * *p* = 0.353 0.127 **	25 (83.3)	13 (61.9)	χ^2^ = 2.957 * *p* = 0.086 0.242 **
No Cardiovascular Risk added	14 (13.9)	3 (17.6)	10 (29.4)		5 (16.7)	8 (38.1)	

^a^ >200 mg/dL. ^b^ <200 mg/dL. ^c^ >150 mg/dL. ^d^ <150 mg/dL. ^e^ <40 years. ^f^ >40 years. * Chi-squared. ** Effect size: Phi and Cramer´s V.

## Data Availability

Datasets generated and analyzed during the current study are available from the corresponding author upon reasonable request.

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
