# Peer review of "Cardiometabolic Risk in a University Community: An Observational Study"

_healthcare, 2024, doi:10.3390/healthcare12171756_

Round 1

Reviewer 1 Report

Comments and Suggestions for Authors

Reviewer 2 Report

Comments and Suggestions for Authors

 This study tried to identify a Cardiometabolic Profile in Teaching, Research, and Service 2 University Personnel. This is an application of a well-known tool at his level and the answer is for a local interest, not to be published in a Q2 Journal.  

The topic is relevant, but when is used to compare different populations, and different individuals. This study did not address a specific gap in this field.

All universities could characterize it's personal (or by its personal), but this is not an interesting aspect for all readers. This is a study of 2019 year published in 2024, before pandemic events (A cross-sectional study was conducted during the 2018-2019 academic year by the Nursing Department, using a self-administered questionnaire and face-to-face assessments of anthropometric variables related to cardiovascular risk in university personnel.)  

The methodology is a simple, a very simple tool for this study, maybe part of a license exam, not for a research article. This article could be more than that.

The conclusions are consistent, but the significance is very low for general academic interest. Generally speaking, the authors could improve the conclusion of their article using a consistent research (more universities, more categories of personal in different area).

References are appropriate.

The references are carefully chosen, from impact journals. They are updated, from the last 10-12 years, in this research area. They constitute a real support for the text of the article.

Figures and are very easy to understand, comparing the lots integrated into this cross-sectional study. They include mean, standard deviation, and statistical significance. I am very simple and detailed at the same time, supporting the conclusion of the action of the investigated population, but is too simple. The article is for students.

Comments on the Quality of English Language

a good English, easy to understand. 

Round 2

Reviewer 1 Report

Comments and Suggestions for Authors

The comments are in the attached file.

Reviewer 2 Report

Comments and Suggestions for Authors

Thank you for all your responses. Great job!

The article is improved, using all recommendations. 

The conclusion should be addressed to this University, not in general. "Women and university academics in health sciences exhibit less healthy lifestyles, (838) characterized by higher prevalence rates of smoking, sedentary behavior, overweight, and (839) abdominal obesity."
